# Rover Attitude and Camera Parameter: Rock Measurements on Mars Surface Based on Rover Attitude and Camera Parameter for Tianwen-1 Mission

Dian Zheng [1], Linhui Wei [1], Weikun Lv [1], Yu Liu [1,2,*] and Yumei Wang [1]

1   School of Artificial Intelligence, Beijing University of Posts and Telecommunications, Beijing 100876, China; zhengdian@bupt.edu.cn (D.Z.); weilinhui@bupt.edu.cn (L.W.); lwk5998@bupt.edu.cn (W.L.); ymwang@bupt.edu.cn (Y.W.)
2   Research Center of Networks and Communications, Peng Cheng Laboratory, Shenzhen 518000, China
*   Correspondence: liuy@bupt.edu.cn

**Abstract:** Rocks, prominent features on the surface of Mars, are a primary focus of Mars exploration missions. The accuracy of recognizing rock information, including size and position, deeply affects the path planning for rovers on Mars and the geological exploration of Mars. In this paper, we present a rock measurement method for the Mars surface based on a Rover Attitude and Camera Parameter (RACP). We analyze the imaging process of the Navigation and Terrain Camera (NaTeCam) on the Zhurong rover, which involves utilizing a semi-spherical model (SSM) to characterize the camera's attitude, a projection model (PM) to connect the image data with the three-dimensional (3D) environment, and then estimating the distance and size of rocks. We conduct a test on NaTeCam images and find that the method is effective in measuring the distance and size to Martian rocks and identifying rocks at specific locations. Furthermore, an analysis of the impact of uncertain factors is conducted. The proposed RACP method offers a reliable solution for automatically analyzing the rocks on Mars, which provides a possible solution for the route planning in similar tasks.

**Keywords:** Mars; rock measure; rover attitude; camera parameter; Navigation and Terrain Camera





## 1. Introduction

Rocks are a significant feature of the surface on Mars and have attracted the attention of Mars exploration efforts as science targets. Martian rocks can pose risks to landers and rovers [1]. The distance, size, and distribution of rocks on Mars can also affect the location and landing of landers [2,3], as well as the choice of routes and autonomous driving of Mars rovers [4]. Rocks can potentially cause abrasion or become temporarily stuck in the wheels of Mars rovers, leading to the failure of exploration missions. For example, Perseverance became trapped on a rock in February 2022 [5]. Accurately measuring the size and distribution of rocks can help understand the information collected from rocks and can provide a deeper understanding of the geological history of Mars [6–8].

There are two main sources of data for rock detection: remote sensing data from orbiting satellites [9] and surface data captured by landers or rovers [10,11]. Remote sensing data are typically collected by cameras on orbiters, such as the High-Resolution Imaging Camera (HiRIC) [12] and Moderate Resolution Imaging Camera (MoRIC) [13] on the Tianwen-1 mission's. High Resolution Imaging Science Experiment (HiRISE), which is carried on the Mars Reconnaissance Orbiter (MRO). Remote sensing data can be used to locate large rocks, the density of craters, and rock abundance [14,15], and it can be further utilized to evaluate the landing safety of a mission [16,17]. They are also regarded as a reference for studying the geological and topographical features of Mars [18]. Surface data are typically collected using scientific and engineering instruments on rovers [19], such as the Navigation and Terrain Camera (NaTeCam) [20] on Zhurong and Hazard Avoidance

Camera (Hazcam) [21] and Supercam [22] on Perseverance. Taking the NaTeCam as an example, it is mounted on the Zhurong rover's mast and serves as a stereo binocular camera primarily used for navigation and guidance.

However, the resolution of orbital remote sensing data is lower than lander data and remote sensing data are restricted to detecting large rocks, limiting their detection capabilities. The NaTeCam can gather surface data about Mars at a significantly higher resolution than remote sensing data. However, each image's relatively limited coverage area results in a reduced amount of information captured per image. The orientation angles and position of the NaTeCam are adjusted based on mission requirements [23]. Therefore, compared to remote sensing data, an approach utilizing the NaTeCam typically requires establishing the relationship between the camera position and orientation before constructing image frames. This process allows for determining rock positions in a unified coordinate system. It is still challenging to accurately identify, measure, and locate rocks in large areas.

The key information of rocks to be measured on Mars includes their size, location on the planet's surface, and distance from Mars rovers. The current method can build the position of the rocks and construct the 3D information of rocks, including their location, orientation, and size. Existing techniques can be enhanced, particularly in scenarios where rocks closely resemble the background in terms of color, there are shadow effects due to lighting conditions, and there are challenges posed by long-distance imaging. Some sophisticated cruise control systems that can be used in real-time can help vehicles drive themselves, but these systems will not work on Mars [24]. Accurate measurements of rocks can be utilized for route planning, even in the Tianwen-2 mission.

The rock measurements include distance and size measurements. Disparity matching based on binocular stereo vision is a practical and effective approach. Estimating stereo disparity is a crucial strategy for reconstructing Mars and has various applications, such as range estimation and terrain mapping [25]. Over the years, several disparity estimation approaches have been proposed. Semi-Global Matching (SGM) [26] is a widely used and popular stereo algorithm that approximates the energy optimization problem by accumulating the cost along several scanlines, reducing the processing time. Semi-Global Block Matching (SGBM) [27] adjusts the mutual information cost module in SGM and achieves good results. Several methods for calculating disparity based on SGM have been developed [28–30]. In [31], a complete matching method based on virtual stereo vision is presented, which makes a 3D reconstruction of objects of varying sizes, such as bubbles. Ref. [32] proposed a 3D reconstruction method based on multi-view cameras, which requires at least three cameras. While this method is viable for processing rocks in stereo images, it does not capitalize on the repetitive information of stones in multiple stereo images. Deep learning approaches, such as [33–35], have been proposed as end-to-end pipeline solutions. They are only appropriate for scenarios related to Mars with prior knowledge, as they require training on already obtained data.

This paper presents a rock measurement method for the Mars surface based on a Rover Attitude and Camera Parameter (RACP). The framework for the RACP is depicted in Figure 1. We analyze the internal relationship of real-world datasets from the NaTeCam in the Tianwen-1 mission. Based on the structure of the camera, the semi-spherical model (SSM) is proposed to describe the attitude of the camera and rotation angles to calculate the position relationship between the camera at different times. Then, the projection model (PM) is developed to calculate the correspondence between data and the Martian scene to restore rock location information in large-scale Martian scenes. The presented method for determining the location and size of Martian rocks is based on Rover Attitude and Camera Parameters, which comprise attitude calculation, projection, disparity matching, filtering, and other components. Experiments on the NaTeCam data in Tianwen-1 demonstrate the effectiveness of the proposed method. The current design framework of Mars rovers and the form of navigation cameras are typically similar. Our method may provide potential insights for similar tasks.

The main contributions of this paper are summarized as follows:

- A semi-spherical model and projection model are introduced to describe the relationship between the rover's attitude, picture data, and the surface of Mars. The regular raster images are expressed in the Mars coordinate system based on the camera specifications and the mechanical structure of the rover.
- We develop an RACP for measuring the distance and size of Martian rocks, combining an SSM and PM. In the SSM and PM, the rotation matrix, translation vector, and other parameters between images are calculated, and the disparity matching method utilizes these parameters.
- The experimental results show that the method we propose can effectively and accurately measure the distance and size of Martian rocks. The results include the distance, size, and location distribution of Martian rocks, based on NaTeCam images.

The remainder of the paper is organized as follows: Section 2 details the proposed semi-spherical model and projection mapping model, and distance and size measuring methods are provided. Section 3 gives the experimental results and discussion. In Section 4, we discuss the impact of uncertain factors. In Section 5, the conclusion is drawn.

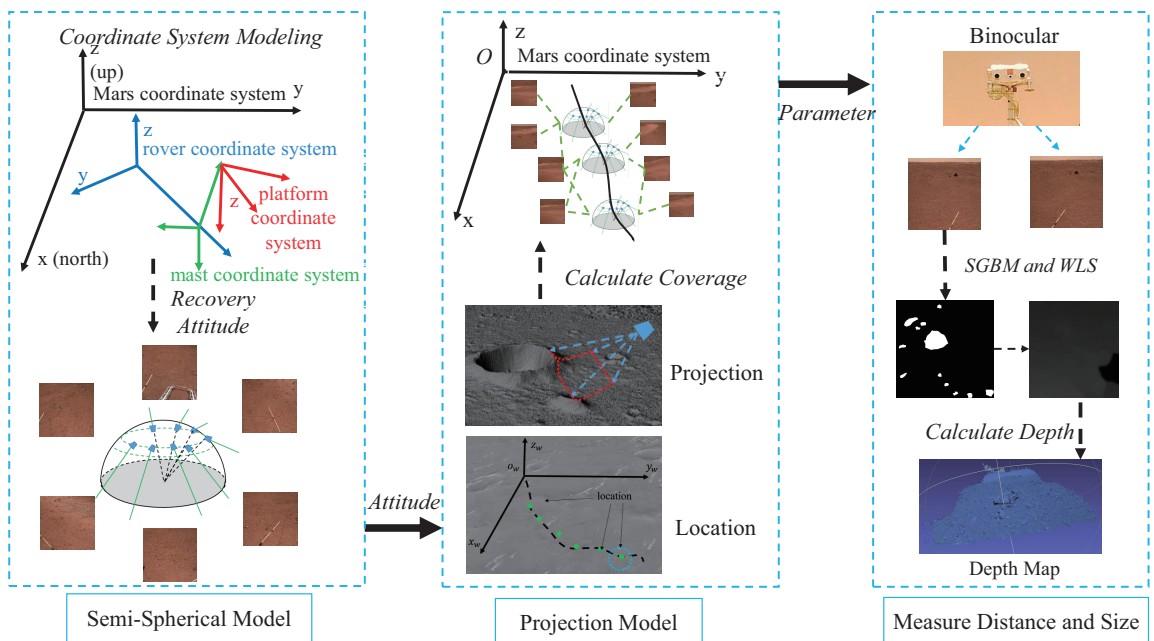

**Figure 1.** The framework of RACP is based on SSM and PM. The SSM and PM extract the distribution range of image data on the Martian surface through a series of operations, including attitude recovery, positioning, projection, and coverage calculation. Rock measurement is based on stereo data and involves operations such as disparity calculation and filtering based on camera parameters.

## 2. Materials and Methods

### 2.1. Zhurong Rover Structure and NaTeCam

On 15 May 2021, China's Zhurong rover successfully landed on the Southern Utopia Plain of Mars [36]. Zhurong is equipped with the NaTeCam, which is crucial for the rover's operations and gathers a vast amount of data about the surface, including information about rocks, dunes, and soil [37]. The NaTeCam comprises two cameras separated by a 27 cm baseline. Each camera has a depth of field ranging from 0.5 m to infinity, with the best focusing at 1 m. The camera has a field of view of $46.5° \times 46.5°$ which allows for a good stereo range resolution up to 30 m from the rover. It has an effective number of pixels of $2048 \times 2048$, and the pixel size is 5.5 micrometers [20]. Liang et al. [19] provide a description of the design, calibration, and performance of the NaTeCam. Zou et al. [20]

describe the physical and functional, performance and operational design of the NaTeCam. Tan et al. [38] discusses the entire process of data product generation, from the design of the data pipeline to data validation.

The NaTeCam is mounted on the Zhurong rover using a mast and a camera platform. The mast of the rover is fixed in length and is attached to the base of the rover. It is able to adjust its yaw and roll angles. The platform is mounted on the mast and can adjust its pitch angle [39]. It is worth noting that the mast and platform are not retractable and can only be turned. During the cruise of the Zhurong rover, the NaTeCam is designed to gather as much information about the terrain of the exploration area as possible. The camera's pointing and acquisition are determined based on the Martian surface conditions and the rover's navigational needs [40]. The NaTeCam employs irregular, selective sampling to adapt to the complex terrain and diverse task requirements of the Mars rover's surface exploration. This makes it challenging to consistently and accurately characterize the camera's attitude.

In this paper, we discuss the mechanical structure and parameters of cameras in Mars rovers and their application in measuring Martian rocks in large-scale Martian scenes. The proposed method is efficient and effective in measuring the distribution of Martian rocks.

### 2.2. Semi-Spherical Model for Zhurong Rover

The camera on the rover acquires images of Mars through irregular sampling. Due to the Martian surface's complexity and the exploration mission's requirements, the angles and positions associated with each NaTeCam image vary. The shooting angles change as the rover moves across the Martian terrain. The relative motion of the camera platform, mast, and rover can modify the camera angle. When investigating the surface of Mars, the approximate region of the image is determined by the NaTeCam's location. The motion of the camera is divided into rotation and translation in the Mars coordinate system, the rover coordinate system, the mast coordinate system, and the platform coordinate system.

In the SSM, the Mars coordinate system and the rover body coordinate system are used to calculate the present coordinates and posture of the rover. In contrast, the mast coordinate system is used to calculate the mast's angle. During the operation of the Zhurong rover, the mast is initially extended vertically upwards. As shown in Figure 2, to complete the imaging of the scientific detection locations, the platform is tilted up or down, and the mast is then yawed. The current camera's imaging attitude must be determined based on the rover coordinates, mast angle, camera height position, and camera optical axis direction. Based on these procedures, we have developed the SSM model to represent the camera's attitude in the Mars coordinate system.

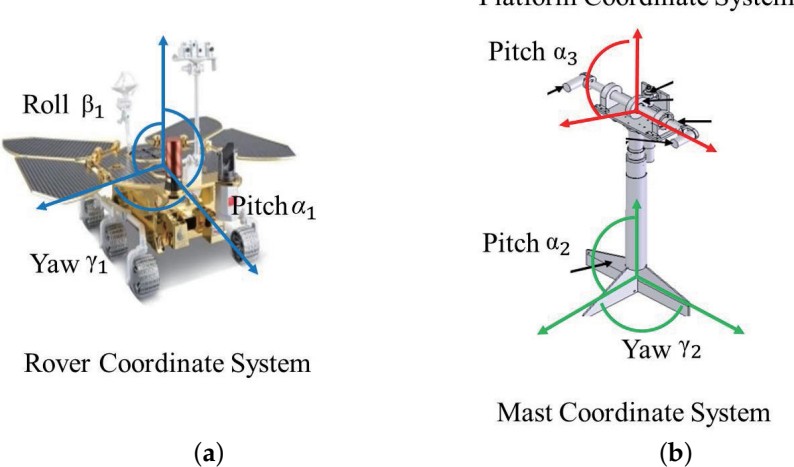

**Figure 2.** The definition of rotation angle in SSM: (**a**) rotation of Zhurong in rover coordinate system; (**b**) rotation of the NaTeCam in mast coordinate system and platform coordinate system.

The $\mathbf{R_x}$, $\mathbf{R_y}$, and $\mathbf{R_z}$ are defined as rotation matrices around the $X$, $Y$, and $Z$ axes, which can be expressed as follows:

$$\mathbf{R_x}(\alpha) = \begin{bmatrix} 1 & 0 & 0 \\ 0 & \cos\alpha & \sin\alpha \\ 0 & -\sin\alpha & \cos\alpha \end{bmatrix} \tag{1}$$

$$\mathbf{R_y}(\beta) = \begin{bmatrix} \cos\beta & 0 & -\sin\beta \\ 0 & 1 & 0 \\ \sin\beta & 0 & \cos\beta \end{bmatrix} \tag{2}$$

$$\mathbf{R_z}(\gamma) = \begin{bmatrix} \cos\gamma & \sin\gamma & 0 \\ -\sin\gamma & \cos\gamma & 0 \\ 0 & 0 & 1 \end{bmatrix} \tag{3}$$

The initial direction of rotation around the $Z$-$Y$-$X$ axis is defined to coincide with the Mars coordinate system. As shown in Figure 2a, we define that the yaw angle around the $Z$-axis is $\gamma_1$, the pitch angle around the $Y$-axis is $\beta_1$, and the roll angle around the $X$-axis is $\alpha_1$. The rotation matrix $\mathbf{R_1}$ can be expressed as follows:

$$\mathbf{R_1} = \mathbf{R_x}(\alpha_1)\mathbf{R_y}(\beta_1)\mathbf{R_z}(\gamma_1) \tag{4}$$

Similarly, the rotation matrix from the rover coordinate system to the mast coordinate system and the rotation matrix from the mast coordinate system to the platform coordinate system can be obtained. As shown in Figure 2b, it is defined that the origin of the mast coordinate system is located at the bottom of the mast and the positive direction of the $Z$-axis is the direction of the mast. The rotation of the mast around the $Z$-axis in the mast coordinate system is known as yaw motion, and the corresponding angle is $\gamma_2$. The rotation around the $X$-axis is pitch motion, and the angle is $\alpha_2$. The rotation matrix $\mathbf{R_2}$ can be expressed as follows:

$$\mathbf{R_2} = \mathbf{R_x}(\alpha_2)\mathbf{R_z}(\gamma_2) \tag{5}$$

The center of the platform is defined as the origin of the platform coordinate system, with the optical axis pointing in the positive direction of the $Z$-axis. In the mast coordinate system, pitch motion is defined as the rotation of the mast about the $X$-axis, and the angle of this rotation is represented by $\alpha_3$. The corresponding rotation matrix $\mathbf{R_3}$ can be expressed as follows:

$$\mathbf{R_3} = \mathbf{R_x}(\alpha_3) \tag{6}$$

Rotation matrix $\mathbf{R}$ can be constructed by combining the matrices $\mathbf{R_1}$, $\mathbf{R_2}$, and $\mathbf{R_3}$:

$$\mathbf{R} = \mathbf{R_1}\mathbf{R_2}\mathbf{R_3} \tag{7}$$

The position and orientation of the NaTeCam in the Mars coordinate system can be determined by combining the odometer and size information of the rover. The data are in PDS4 standards and can be found in the label of the data [38]. This information is necessary for accurately determining the location and orientation of the camera in relation to the surface of Mars.

*2.3. Projection Model*

Recovering the topography of Mars is a significant scientific research mission and a crucial foundation for the autonomous driving of rovers. This can be achieved by acquiring information about the surrounding terrain and other surroundings through the use of cameras. Determining the approximate range of a camera shot is a fundamental task in this process. To capture photographs of the Martian surface, the camera's position relative to

the rover may be altered at the same location, or the rover may be moved to other locations. To achieve this, it is necessary to adjust the position of both the rover and the camera.

In this paper, projection is used to approximately estimate the area of the camera shot, as shown in Figure 3. In most cases, there will be a rotation angle between the rover and Mars, which can be established by the SSM. The camera's translation motion consists of two translation vectors in the Mars coordinate system, with the landing location as the origin of the coordinates. $\mathbf{t_1}$ is the vector from the origin of the Mars coordinate system to the center of the rover, and $\mathbf{t_2}$ is the vector from the origin of the rover coordinate system to the center of the camera. The rotation matrix from the rover coordinate system relative to the Mars coordinate system is $\mathbf{R_1}$. The translation vector $\mathbf{t}$ from the origin of the Mars coordinate system to the center of the camera is as follows:

$$\mathbf{t} = \mathbf{t_1} + \mathbf{R_1}\mathbf{t_2} \tag{8}$$

where $\mathbf{R}$ and $\mathbf{t}$ are the attitude angle and location of the camera. Moreover, the PM can estimate the image's range roughly.

The NaTeCam's imaging model is the pinhole model, which specifies how points are recorded in the picture. Figure 4 demonstrates the imaging principle of the pinhole camera. Based on the pinhole imaging paradigm, the PM can be divided into three coordinate system conversions. As shown in Figure 4, the pixel coordinate system is transformed into the picture coordinate system. The point $(u_0, v_0)$ represents the principal point in the center of the image, and $(d_x, d_y)$ denotes the size of the pixels on the photosensitive element of the NaTeCam. The point $(u, v)$ represents the pixel coordinate position corresponding to the target point $P$, and $(u_i, v_i)$ represents its corresponding location in the image coordinate system. The equation for the projection from the pixel coordinate system to the image coordinate system is as follows:

$$\begin{bmatrix} x_i \\ y_i \\ 1 \end{bmatrix} = \begin{bmatrix} d_x & 0 & -u_0 d_x \\ 0 & d_y & -v_0 d_y \\ 0 & 0 & 1 \end{bmatrix} \begin{bmatrix} u \\ v \\ 1 \end{bmatrix} \tag{9}$$

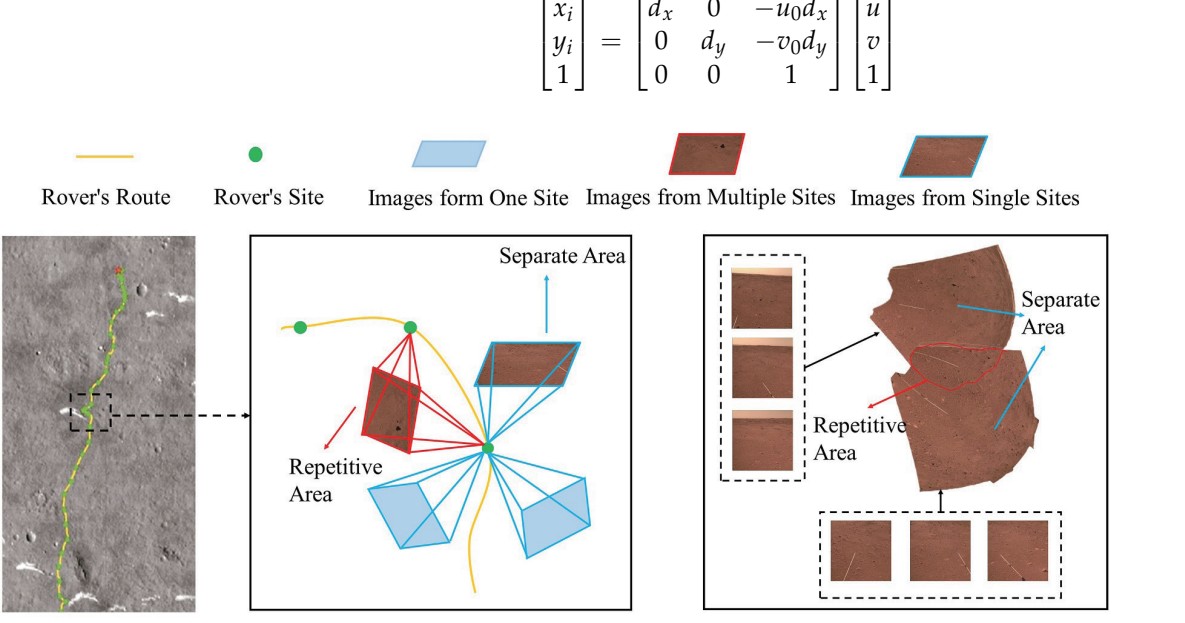

(a) Rover Exploration Area  (b) Projection of Local Images  (c) Result of Projection

**Figure 3.** The framework of PM. (**a**) is a map of the area explored by the Mars rover. (**b**) illustrates the projection process of images in a local area. The area captured by the image is divided into a repeat area and separate area. The repeat area can be captured at multiple filming locations, while the separate area can only be captured at a single location. (**c**) is the result of projection.

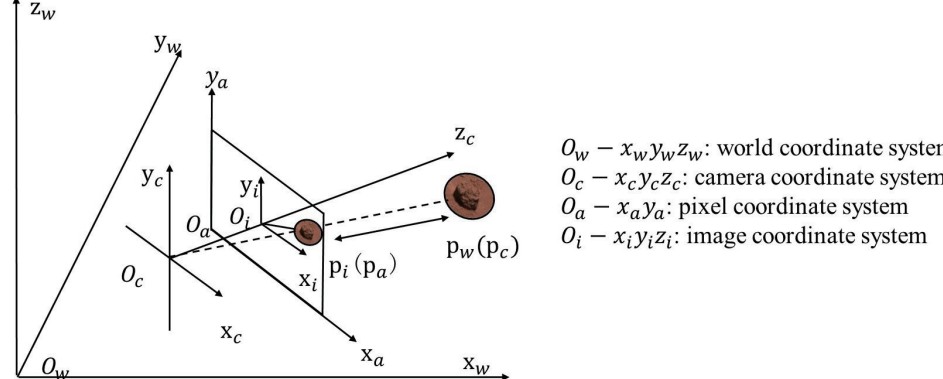

$O_w - x_w y_w z_w$: world coordinate system
$O_c - x_c y_c z_c$: camera coordinate system
$O_a - x_a y_a$: pixel coordinate system
$O_i - x_i y_i z_i$: image coordinate system

**Figure 4.** The pixel coordinate system is in the same plane as the image coordinate system. The angle between the camera coordinate system and the Mars coordinate system can be obtained according to the SSM.

The second transformation converts the picture coordinate system to the camera coordinate system. $(x_c, y_c, z_c)$ is the coordinates of point $P$ in the camera coordinate system and $f$ is the focal length of the NaTeCam. The conversion from the image coordinate system to the camera coordinate system can be represented by the following equation:

$$\begin{bmatrix} x_c \\ y_c \\ z_c \end{bmatrix} = \begin{bmatrix} \frac{z_c}{f} & 0 & 0 \\ 0 & \frac{z_c}{f} & 0 \\ 0 & 0 & z_c \end{bmatrix} \begin{bmatrix} x_i \\ y_i \\ 1 \end{bmatrix} \tag{10}$$

The third transformation is from the camera coordinate system to the world coordinate system. This process requires the rotation matrix **R** and translation vector **t**, which can be determined using SSM. The equation for this transformation is as follows:

$$\begin{bmatrix} x_w \\ y_w \\ z_w \\ 1 \end{bmatrix} = \begin{bmatrix} \mathbf{R} & \mathbf{t} \\ \mathbf{0}^T & 1 \end{bmatrix}^{-1} \begin{bmatrix} \frac{d_x z_c}{f} & 0 & \frac{-u_0 d_x z_c}{f} & 0 \\ 0 & \frac{d_y z_c}{f} & \frac{-v_0 d_y z_c}{f} & 0 \\ 0 & 0 & z_c & 0 \\ 0 & 0 & 0 & 1 \end{bmatrix} \begin{bmatrix} u \\ v \\ 1 \\ 1 \end{bmatrix} \tag{11}$$

It is important to note that the coverage of the image provided by the PM is approximate, and the results will be refined in Section 4 using stereo matching. Some distant points may be incorrect due to the limitations of current feature extraction techniques, such as SIFT [41], SURF [42], and ORB [43]. Additionally, pixels may also be discarded if they cannot be matched, as is often the case with pixels at the horizon or in the sky.

### 2.4. Measuring Distance Based on PM

Measuring the distance between the rock and the rover is an essential prerequisite for choosing a route. The steps for distance measurement using the binocular vision system are illustrated in Figure 5. First, the internal and external parameters of the camera are calculated by the PM, which plays a crucial role in creating the disparity map. Another role of the PM is to categorize rocks into two distinct groups: repeatable rocks and single rocks. Then, the picture distortion is calibrated and rectified, aligning the left and right images on the same plane. The steps encompass undistortion, rectification, and epipolar constraint. The rectified stereo images are employed to estimate disparity, a process for which we utilize SGBM. Finally, the disparity and depth maps are calculated based on the matching feature points to determine the distance to the target rocks. In the case of repeatable rocks, the final measurement result can be determined through weighted calculations across multiple angles.

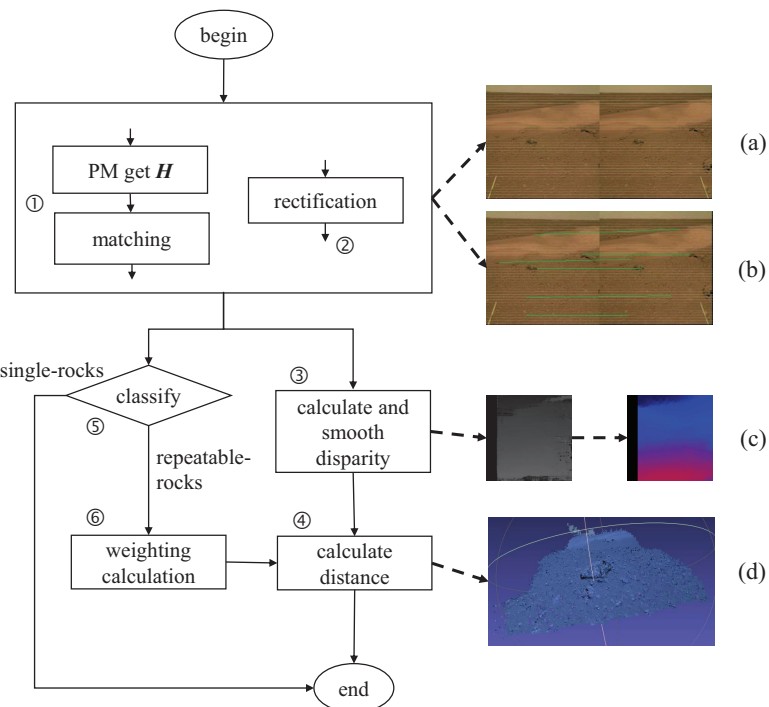

**Figure 5.** The framework of measurement. (**a**) is a rectified stereo image, in which the corresponding pixels are on the same line. (**b**) shows the matching of feature points. (**c**) is the obtained disparity map, which has undergone color correction. (**d**) shows the spatial position of the obtained rock portion

(1) The internal parameters, external parameters, rotation matrix **R**, and translation vector **t** of the NaTeCam are necessary for measuring stone distance. In the PM, the rotation matrix **R** and translation vector **t** are calculated. Then, we calculate homography matrix **H** for two images collected between the left and right cameras simultaneously. We calculate the homography matrix **H** more accurately based on camera parameters [44]. The rotation matrix **R** and translation vector **t** obtained by the SSM are used to calculate **H**. The $s$ is a scale factor, $r_{ij}$ $(i, j = 0, 1, 2)$ is the component of **R**, and $t_i$ $(i = 0, 1, 2)$ is the component of **t**. **H** can be computed as follows:

$$\mathbf{H} = s\mathbf{K} \begin{bmatrix} r_{00} & r_{01} & t_1 \\ r_{10} & r_{11} & t_2 \\ r_{20} & r_{21} & t_3 \end{bmatrix} \tag{12}$$

where **K** is the camera intrinsics matrix, defined as follows:

$$\mathbf{K} = \begin{bmatrix} f_x & 0 & c_x \\ 0 & f_y & c_y \\ 0 & 0 & 1 \end{bmatrix} \tag{13}$$

$f_x$ and $f_y$ represent the focal lengths in the image coordinate system, and $c_x$ and $c_y$ denote the principal point's image coordinates. This matrix encapsulates the camera's internal parameters, including focal lengths, principal point coordinates, and image scaling. Feature points of rocks in overlapping regions are detected and matched, and the results can be used for classification.

(2) Achieving perfect forward alignment between the left and right cameras can be challenging. To coplanarize the pictures and precisely align the image pixels between the left and right cameras, the translation vector and rotation matrix must be determined using the PM. Before stereo rectification, the picture distortion in the image pairings is severe, and

the corresponding points are not on the same line. However, after stereo rectification, the image distortion is corrected, and the corresponding points are essentially on the same line.

(3) The matched binocular camera images can be used to estimate the disparity map, as illustrated in Figure 6. To adjust the data, the centers between the left and right cameras, $O_{left}$ and $O_{right}$, are positioned on the same horizontal line, with the distance between them known as the baseline $b$. Both the left and right images are placed within the $O_w - X_w Y_w Z_w$ coordinate system. The optical axes between the left and right cameras, $Z_{left}$ and $Z_{right}$, are parallel to each other. Mapping points $P$ on the image planes between the left and right cameras are represented by $P_{left}$ and $P_{right}$, respectively. The disparity, $d$, is the difference between the column coordinates of $P_{left}$ and $P_{right}$. In this paper, the images corrected by the PM are used as input data for the SGBM [27] algorithm to determine $d$ based on the binocular camera data. The steps encompass cost computation, cost aggregation, path optimization, disparity selection, and subpixel refinement.

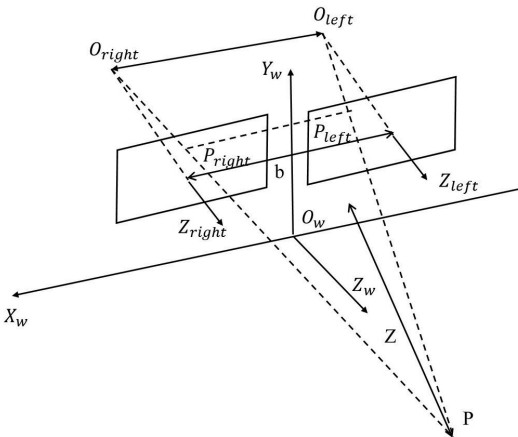

**Figure 6.** The imaging principle of binocular camera.

(4) After image rectification, the forward parallel alignment of the camera is accurate, meaning that each pixel line in the left camera is aligned with each pixel line in the right camera. The disparity $d$ between the left and right photos is inversely proportional to the depth $Z$. Based on the principle of triangle similarity, the following relationship can be established:

$$Z = \frac{fb}{d} \tag{14}$$

Based on the disparity, baseline $b$, and focal length $f$, the depth $Z$ from the rock to the camera can be calculated. $Z$ is the distance between pixels and the center of the NaTeCam. In PM, the rotation matrix **R** and translation vector **t** are obtained, so the position of the pixel can then be calculated, as shown in Figure 6.

Based on the position of the optical center and the direction of the optical axis, the coordinates of the pixels in the 3D world can be calculated.

(5) There are two types of rocks: repeatable rocks and single rocks. In Figure 7, rocks A and B are repeatable rocks and rocks C–F are single rocks. Figure 8 shows the measuring principle diagram for detecting repeatable rocks. Repeatable rocks can be measured at several locations and the results can be independently validated by calculating the distance and angle of the rock relative to each site. However, some prominent rocks may only appear once in the dataset. Additionally, some pebbles may appear multiple times, but the distance between them is too great for mutual verification. In these cases, we selected single rocks with high accuracy.

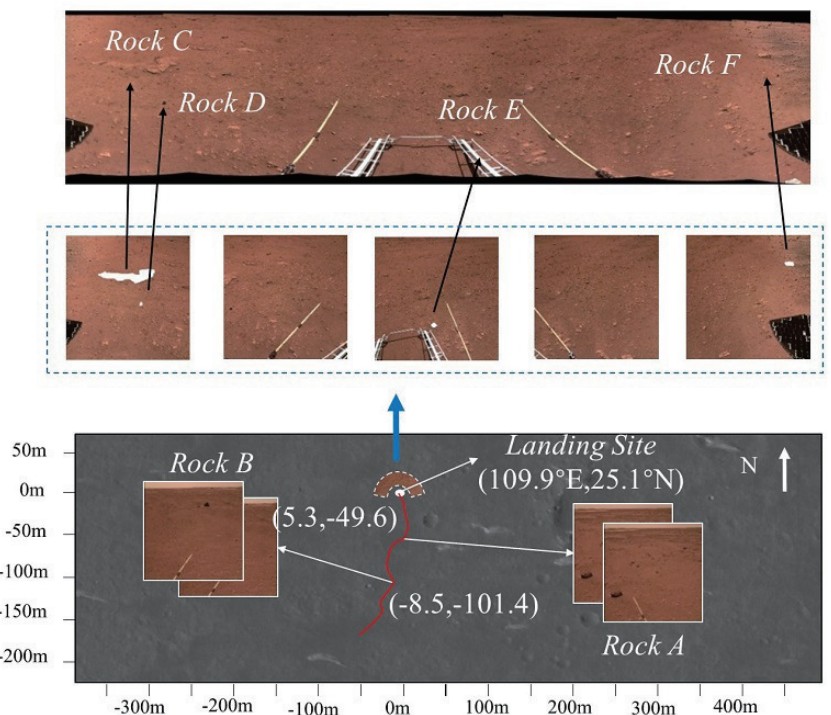

**Figure 7.** Classification and distribution of rocks. Rocks A and B appear in multiple stations (9 June 2021 and 13 June 2021), and rocks C–F appear in the landing site. Rocks C–F are photographed at the landing site of Zhurong (109.9°E, 25.1°N) (18 May 2021). In the Mars coordinate system of Mars with the landing site as the origin, rock a is photographed at (−8.5 m, −101.4 m), and rock B is photographed at (5.3 m, −49.6 m).

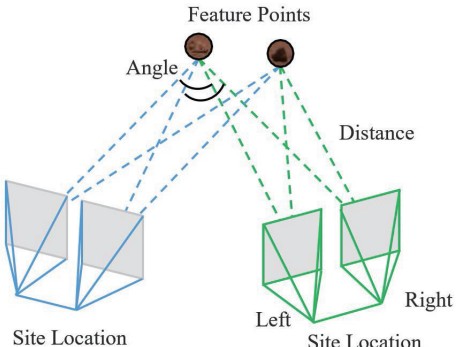

**Figure 8.** The location of repeated rocks can be verified based on the calculation results from different sites.

(6) Repeatable rocks allow for the verification of their positional information from various angles. When calculating the rock positions, the distances between the rocks and the camera are weighted. The closer the position, the higher the credibility of the value obtained. If a rock has measurement results from $N$ angles, corresponding to rock positions $P_i$ and distances $d_i$ from the camera, $d_{max}$ is the maximum distance among all distances. The final rock position $P$ can be represented as follows:

$$P = \sum_{i \in N} P_i \frac{d_{max} - d_i}{\sum_{i \in N} d_i} \tag{15}$$

*2.5. Measuring Size Based on PM*

Obtaining information on the size of Martian rocks can prevent the Mars rover from getting stuck while in motion. The size of the rocks can also indicate past geological activity on the Martian surface, such as water flow and weathering. Measuring rock size using stereo image-based methods includes identifying rock targets, segmenting rock contours, and calculating contour positions.

The recognition and segmentation technique is essential for determining the size of rocks [45]. We have developed the MarsNet [46] for Martian rock detection and released the Tianwen-1 rock detection dataset, which is available at https://github.com/BUPT-ANT-10 07/MarsNet (accessed on 17 April 2022). In this paper, we used MarsNet to identify and segment Mars rocks and measure their size based on the SSM and PM.

To compute the depth of a pixel corresponding to a specific point, we use stereo matching to construct a disparity map and apply Equation (13). However, the quality of the disparity map directly obtained through this method could be better due to repetitive textures, weak textures, overexposure, and noise. In many Martian landscapes, the color of the rocks and soil is similar, leading to false disparity results for rock edges. As a result, the contour of the rock could be more consistent between the disparity map and the effects of semantic segmentation. Therefore, it is necessary to apply filters to the disparity map.

In this paper, we use the weighted least squares (WLS) filter [47] to filter the disparity map. The WLS filter is a type of filter that is used to reduce the noise and errors present in the disparity map. The WLS filter is an edge-preserving filter that aims to make the filtered output as close as possible to the original image, smooth out small gradient areas, and preserve the edges of strong gradients. It is based on the least squares criterion, which minimizes the sum of squared errors between the estimated and true disparities. This process converts sparse disparity into dense disparity and reduces noise in the map, as shown in Figure 9. We use the results of semantic segmentation to identify the rock's contour and validate it using the disparity map. When they correspond, we use the disparity map to calculate the position of the pixel on the rock edge and further obtain the size of the rock.

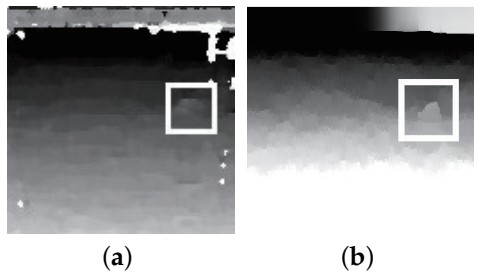

(**a**)　　　　　　　　　(**b**)

**Figure 9.** WLS filters the disparity map to preserve the effect of rock edge: (**a**) disparity map; (**b**) disparity map with WLS.

## 3. Results

*3.1. NaTeCam Datasets from Tianwen-1 Mission*

The data utilized in this study were captured by the NaTeCam, with acquisition dates including 18 May 2021; 9 June 2021; 13 June 2021; 3 September 2021; and 22 January 2022. The camera's earliest operational date was 18 May 2021, during which data were obtained from the vicinity of the landing site. Subsequently, the Zhurong rover embarked on exploratory activities and recorded data around the roaming route. These datasets primarily encompass two types of rocks: individual rocks and repeatable rocks. The intrinsic parameter matrix of the camera can be obtained by parsing the labels in the data, which includes the focal length, principal point offset, etc. The NaTeCam images used in this study were transformed from the Bayer format to the RGB-per-pixel format at level 2C. Each piece of data contains two paired files: the data file and the corresponding

tag file. These datasets can be obtained from the China National Space Administration (http://moon.bao.ac.cn (accessed on 3 August 2021)).

### 3.2. Result of Measuring Distance

We take pairs of image data from the left and right cameras of the NaTeCam to measure the distance of the rock. The intrinsic matrix is computed using the label file data of the NaTeCam. $\mathbf{M_{left}}$ is the intrinsic matrix of the left camera, as follows:

$$M_{left} = \begin{bmatrix} 2395.309 & 0 & 1025.862 \\ 0 & 2395.309 & 1028.034 \\ 0 & 0 & 1 \end{bmatrix} \tag{16}$$

$\mathbf{M_{right}}$ is the intrinsic matrix of the right camera, as follows:

$$M_{right} = \begin{bmatrix} 2394.445 & 0 & 1022.005 \\ 0 & 2394.445 & 1026.784 \\ 0 & 0 & 1 \end{bmatrix} \tag{17}$$

We calculate the extrinsic parameters of the image data to be detected in the SSM, including the rotation matrix and translation vector. Taking rock A in Figure 7 as an example, we can obtain the following:

$$R_{a1} = \begin{bmatrix} 0.677 & -0.630 & -0.381 \\ 0.705 & 0.703 & 0.088 \\ 0.212 & -0.328 & 0.921 \end{bmatrix} \tag{18}$$

$$t_{a1} = \begin{bmatrix} -3.763 & -56.034 & 0.632 \end{bmatrix} \tag{19}$$

The $\mathbf{R_{a1}}$ represents the rotation matrix between the left cameras and the starting direction, and $\mathbf{t_{a1}}$ represents the translation vector between the left cameras and the initial location.

The matrices of rotation relative to the initial location are $\mathbf{R_{left}}$ and $\mathbf{R_{right}}$, and the translation vectors are $\mathbf{t_{left}}$ and $\mathbf{t_{right}}$. We compute the homography matrix between two pictures based on the PM and the rock distance based on the SGBM algorithm [27]. This step is implemented on opencv.

As shown in Figure 7, rocks A and B are present in multiple locations, while rocks C–F belong to the same location. For rocks A and B, there are multiple SSM models available. By the position and distance of the rocks relative to locations, we verify the accuracy of computations. The results of the camera attitude and rock distance measurements are shown in Table 1. In certain specific scenarios, such as the landing site depicted in Figure 7, rocks C–F only appear in one SSM. We estimated the rock distance around Zhurong's landing site and compared it to the results of Wu et al. [17], as shown in Table 1. The results indicate that the method for estimating rock distance is accurate.

**Table 1.** Measuring distance of single rocks (C–F) and repetitive rocks (A and B).

| Number | Camera Position/m | Pixel Position | Measured Value/m | Wu [17] |
|---|---|---|---|---|
| A1 | [−3.763, −56.040, 0.632] | [90, 342] | 4.37 | \ |
| A2 | [−3.709, −56.110, 0.599] | [103, 364] | 3.92 | \ |
| B1 | [4.596, −50.043, 2.360] | [337, 119] | 17.86 | \ |
| B2 | [4.558, −50.13, 2.358] | [126, 106] | 17.48 | \ |
| C | [0.590 , 0.185, 2.853] | [328, 155] | 9.88 | 10 m |
| D | [0.676, 0.204, 2.853] | [335, 280] | 6.38 | 6 m |
| E | [0.820, 0.112, 2.855] | [276, 368] | 5.21 | 5 m |
| F | [0.746, −0.118, 2.860] | [121, 121] | 10.82 | 10 m |

### 3.3. Result of Measuring Size

The weak texture nature of the Martian surface scene often makes it difficult to distinguish rocks and backgrounds in the disparity map. To address this issue, we selected the SGBM [27] matching pattern and WLS filtering methods. We tested different pairs of images, and the results are shown in Figure 10.

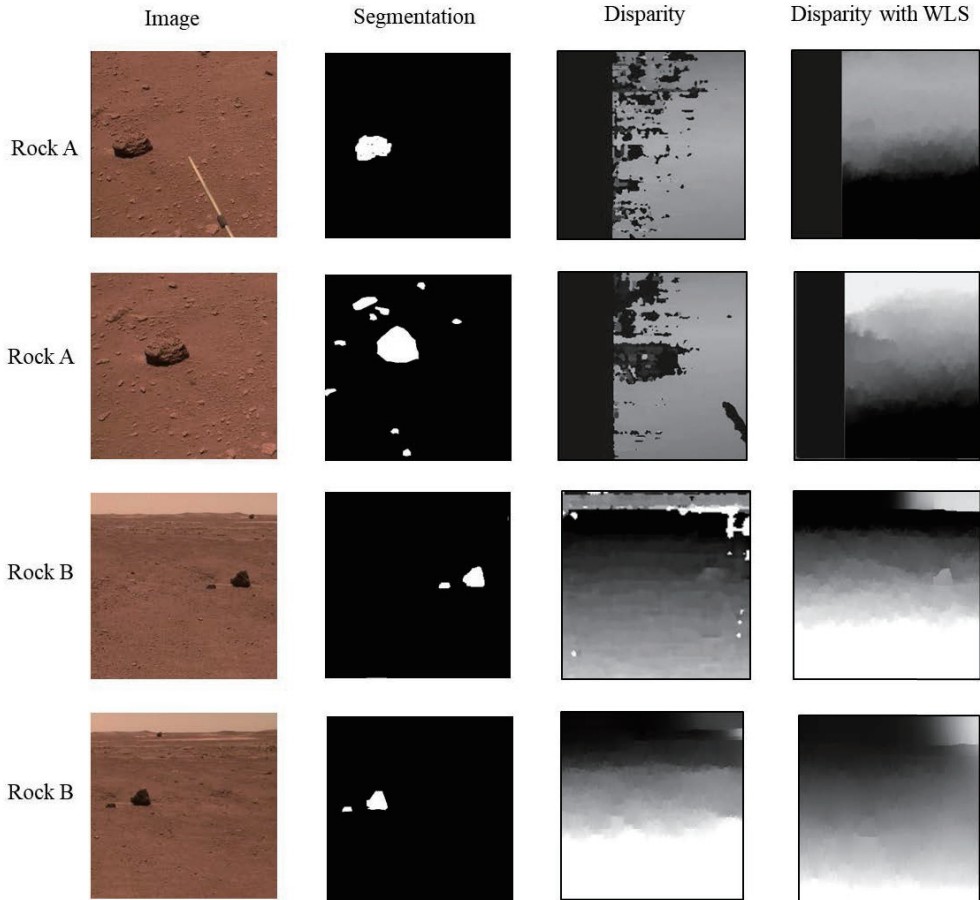

**Figure 10.** The result of disparity estimation. The first column is the input image, and the second column is the semantic segmentation result corresponding to the rock. The third column is the unfiltered disparity map. The fourth column is the filtered disparity map. The data were collected on 9 June 2021 and 13 June 2021.

To calculate the size of the rock, we selected pixels on the contour of the rock, located on the left, right, top, and bottom of the rock. Using the depth map obtained through disparity matching, we calculated the position of these four pixels in the camera coordinate system, as shown in Table 2. We then computed a depth map from the disparity map to recover the coordinates of each pixel in the camera coordinate system. To determine the rock size, the pixel point at the extreme value of the rock's contour was selected. The horizontal and vertical dimensions of the rock are calculated.

When the texture and brightness of the rock differ from the soil, the disparity map will reveal the contour of the rock. In some datasets, the textures of rocks and soil are so similar that it is impossible to detect the rocks on the disparity map, as seen in Figure 11b. To accurately determine the size of the rock in these cases, we enhance the filter, which has the advantage of showing how the rock's dimensions change continuously.

We assess the size of rocks around the landing site and compare them to the findings of Wu et al. [17], as shown in Table 2, and the results are consistent with expectations.

**Table 2.** Size of the rocks around the landing site.

| Rock | Point | Index | Position/m | Size/m | Wu [17] |
|------|-------|-------|------------|--------|---------|
| C | left | [150, 162] | [−1.400, −1.429, 9.024] | 3.47 | 3.2 m |
|   | right | [380, 150] | [2.070, −1.784, 9.999] |  |  |
|   | top | [329, 137] | [1.284, −2.111, 1.055] | 1.13 |  |
|   | bottom | [328, 187] | [1.010, −0.981, 8.416] |  |  |
| D | left | [342, 280] | [0.926, 0.251, 6.459] | 0.11 | 0.15 m |
|   | right | [331, 280] | [0.816, 0.253, 6.526] |  |  |
|   | top | [335, 287] | [0.819, 0.313, 6.218] | 0.14 |  |
|   | bottom | [335, 272] | [0.876, 0.169, 6.648] |  |  |
| E | left | [267, 367] | [0.094, 0.953, 5.175] | 0.18 | 0.18 m |
|   | right | [288, 369] | [0.272, 0.958, 5.112] |  |  |
|   | top | [276, 363] | [0.173, 0.906, 5.218] | 0.11 |  |
|   | bottom | [276, 374] | [0.169, 1.015, 5.080] |  |  |

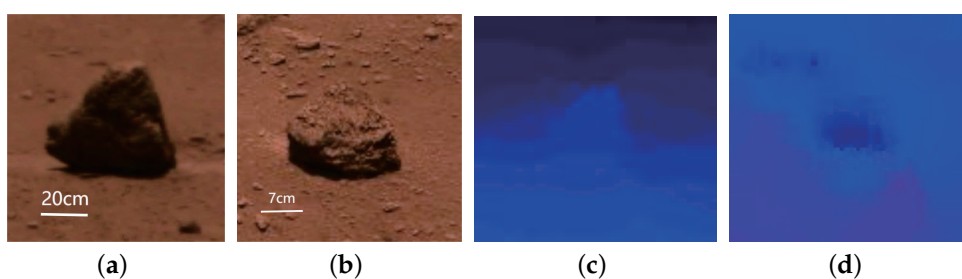

|     (a)     |     (b)     |     (c)     |     (d)     |

**Figure 11.** Some filtering settings are employed. (**a**) shows stones that are clearly distinguishable from the background. (**b**) shows stones that are not clearly distinguishable from the background. (**c**) is the depth map of stone (**a**) with a clear edge. (**d**) is the depth map of stone (**c**) with an unclear edge. The data were collected on 9 June 2021 and 13 June 2021.

### 3.4. Distribution of Rocks

In distance estimation, the repeated texture led to tortuous distances, as illustrated in Figure 12a. We optimized the filtering for this circumstance and obtained a relatively smooth border line, as shown in Figure 12b.

As shown in Figure 13, we measure the locations of rocks captured by Zhurong at two sites where photos were taken on 3 September 2021 and 22 January 2022. The distances from the NaTeCam's location are calculated, and the results are presented in Table 3. In order to reduce inaccuracies, Rock 2, Rock 6, Rock 7, and Rock 11 are used to align different images. Additionally, the distribution of rocks in the panoramic view of Mars is depicted in Figure 14, and the results are consistent with expectations.

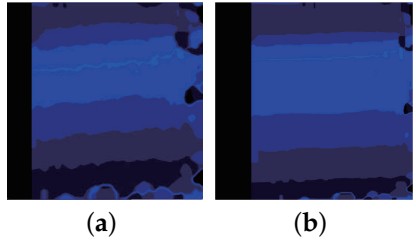

|     (a)     |     (b)     |

**Figure 12.** The filter smooths the distance measurements: (**a**) unfiltered depth map; (**b**) filtered depth map.

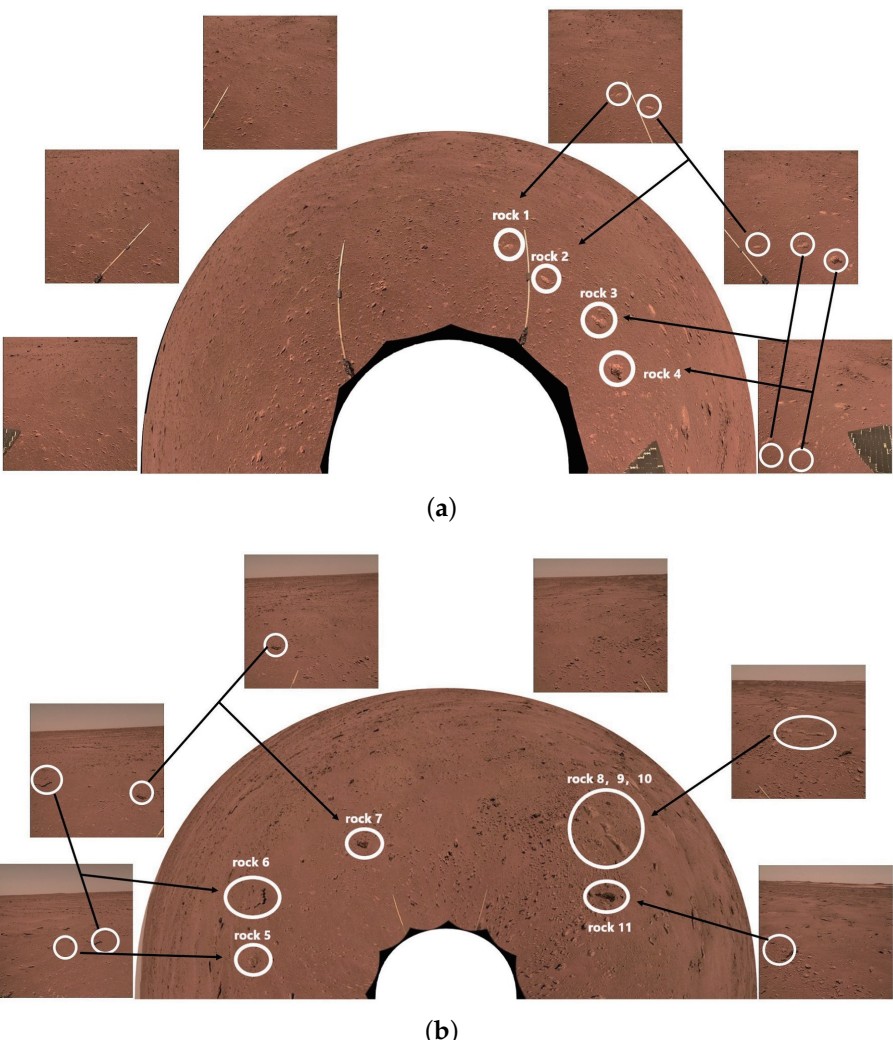

(**a**)

(**b**)

**Figure 13.** Images captured by the NaTeCam onboard the Zhurong on 22 January 2022 and 3 September 2021. Each image is stitched together from six NaTeCam images. Rocks 1–11 are marked by the red circle. (**a**) A 180-degree surround image of the Martian surface on 22 January 2022. (**b**) A 180-degree surround image of the Martian surface on 3 September 2021.

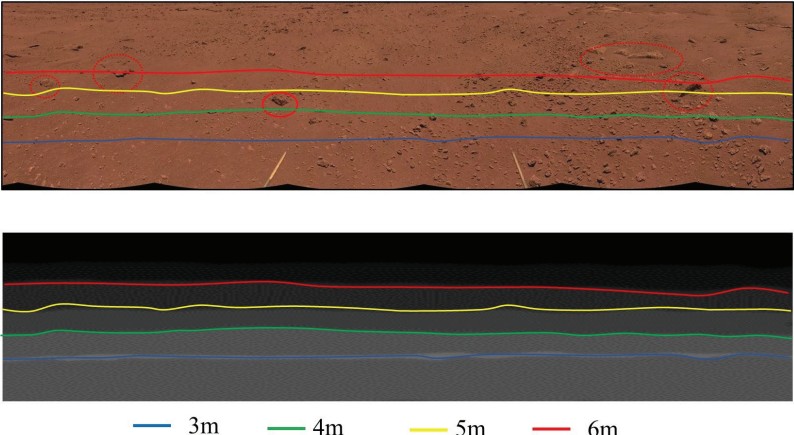

**Figure 14.** Results of the location of Martian rock in Figure 13. The data were collected on 3 September 2021.

**Table 3.** Results of rock distance measurements in Figure 13. Rocks 2–4, 6, 7, and 11 are repeatedly measured.

| Date | Rock Number | Camera Position/m | | | Camera Angle/° | | | Measured Value/m |
|---|---|---|---|---|---|---|---|---|
| | | x | y | z | Roll | Pitch | Yaw | |
| 22 January 2022 | 1 | 143.303 | −1017.045 | 13.540 | 3.844 | −60.034 | −173.100 | 3.30 |
| | 2 | 143.303 | −1017.045 | 13.540 | 3.844 | −60.034 | −173.100 | 2.90 |
| | 2 | 143.252 | −1016.955 | 13.540 | −39.160 | −50.421 | 135.509 | 2.86 |
| | 3 | 143.252 | −1016.955 | 13.540 | −39.160 | −50.421 | 135.509 | 2.94 |
| | 3 | 143.254 | −1016.852 | 13.539 | −56.061 | −28.049 | 109.156 | 2.96 |
| | 4 | 143.252 | −1016.955 | 13.540 | −39.160 | −50.421 | 135.509 | 2.69 |
| | 4 | 143.254 | −1016.852 | 13.539 | −56.061 | −28.049 | 109.156 | 2.73 |
| 3 September 2021 | 5 | −95.672 | −900.315 | −6.800 | 72.512 | 29.125 | −82.262 | 5.25 |
| | 6 | −95.672 | −900.315 | −6.800 | 72.512 | 29.125 | −82.262 | 6.72 |
| | 6 | −95.604 | −900.347 | −6.802 | 73.553 | 0.272 | −90.667 | 6.89 |
| | 7 | −95.604 | −900.347 | −6.802 | 73.553 | 0.272 | −90.667 | 4.89 |
| | 7 | −95.561 | −900.409 | −6.805 | 70.128 | −28.411 | −99.659 | 4.81 |
| | 8 | −95.587 | −900.552 | −6.810 | −1.802 | −72.421 | −179.269 | 7.22 |
| | 9 | −95.587 | −900.552 | −6.810 | −1.802 | −72.421 | −179.269 | 6.83 |
| | 10 | −95.587 | −900.552 | −6.810 | −1.802 | −72.421 | −179.269 | 6.53 |
| | 11 | −95.587 | −900.552 | −6.810 | −1.802 | −72.421 | −179.269 | 5.15 |
| | 11 | −95.649 | −900.595 | −6.811 | −59.565 | −55.335 | 119.393 | 5.10 |

## 4. Discussion

Due to the constraints of the Martian environment, operations such as camera calibration and hand–eye calibration, which are easily performed on Earth, are challenging. This leads to significant errors in 3D measurements in the Martian setting. Our work focuses on rocks around the Mars rover, aiming to measure the distance between rocks and the rover, as well as the size of the rocks, particularly at relatively close distances. Therefore, our method needs to consider three main sources of errors: (1) errors from the Mars rover's mechanical joint movements, (2) errors from the pinhole camera model calibration, and (3) errors from image matching.

### 4.1. Errors from Rover

Errors in the mechanical joint movements of the Mars rover can impact measurement accuracy, with the accuracy decreasing as the distance increases. Ref. [40] provides results for the pointing error of the mast platform: the maximum angular error $\theta$ satisfying the $3\sigma$ criterion is 1.25°. Errors in angular measurements can affect results in two ways: by providing incorrect camera-relative rover position information and by yielding erroneous rock position information based on camera location data. The distance of the camera from the center of the rover is approximately a maximum of 1.8 m. The optimal distance for the NaTeCam camera to operate is from 7 to 8 m, and we assume 10 m for convenience. $d$ represents the sum of the distance from the camera to the rover and the distance from the rock to the camera, and $\theta$ denotes the angle error. Then, the measurement error $\delta$ can be calculated using the following equation:

$$\delta = 2d \sin(\frac{\theta}{2}) \tag{20}$$

Choosing the maximum angle error $\theta$ satisfying the $3\sigma$ principle, and with $d$ being 11.80 m, the maximum measurement error amounts to 257 mm.

The probability density function for the final rock measurement error $\delta$ can be derived as follows:

$$f(\delta) = \frac{4 \arcsin^3 \frac{\delta}{d}}{d\sigma^5 \sqrt{2\pi(1 - \frac{\delta}{d})^2}} e^{-\frac{\arcsin^2 \frac{\delta}{d}}{2\sigma^2}} \tag{21}$$

where $3\sigma$ is 1.25° and $d$ represents the distance between the center of rover and the rock. It can be approximated as the sum of the distance from the camera to the rover and the distance from the rock to the camera.

### 4.2. Errors from Pinhole Model

Utilizing the pinhole model to correct distortion is a factor influencing measurement accuracy, and this process is accomplished through camera calibration. Calibrating the NaTeCam presents a complex challenge, especially within the extraterrestrial environment where comprehensive calibration conditions are not feasible. Typically, calibration is performed prior to the rover's launch. The correction-related work for the NaTeCam was undertaken by [20] before the launch, and it serves as our reference. Their attained outcome revealed a maximum test distance of 10 m, yielding a 3D accuracy of approximately 20 mm.

### 4.3. Errors from Image Matching Error

Matching errors directly affect depth calculation. The impact of matching errors on depth in a stereo system is related to the following parameters: the baseline $B$ of the stereo camera, the pixel size $s_0$, and the spatial dimension of the focal length $f_0$. Given an error in disparity $\Delta d$ under the current disparity $d$, the final depth error $\delta$ can be expressed as follows:

$$\delta = B\frac{f_0}{s_0}\left(\frac{1}{d} - \frac{1}{d - \Delta d}\right) \tag{22}$$

In the case of the NaTeCam used in this paper, the baseline $B$ is approximately 270 mm, the pixel size $s_0$ is 5.5 μm, and the spatial length of the focal length $f_0$ is 13.1 mm. On the experimental dataset with a testing distance of 10 mm, the matching error of each pixel leads to an error of 153 mm in the final depth.

### 5. Conclusions

This paper presents an approach for measuring the distance and size of Martian rocks using the Zhurong rover. The proposed method, the RACP, provides a robust and efficient means of determining the size and distance of multiple rocks in a sizeable Martian scenario. The SSM model describes the relationship between the rover's attitude and the data from the NaTeCam, while the PM method projects the image of the Martian surface onto the Mars coordinate system, obtaining the rotation matrix and translation vector and calculating the homography matrix between the images. This model provides an approach for determining the distance and size of Mars rocks and defines a data modeling technique specifically for the Zhurong rover. We further analyzed the influence of error factors on the measurement results, including rover and camera errors. Overall, this research offers a valuable contribution to Mars exploration and provides a crucial tool for future missions to the red planet.

In the future, further experiments should be conducted to improve the measurement of the size of Martian rocks. This paper only assesses the size of Martian rocks in a two-dimensional plane due to a need for more data from multiple angles. Therefore, methods for estimating the three-dimensional size of rocks should also be explored.

**Author Contributions:** Conceptualization, D.Z.; methodology, D.Z. and L.W.; software, D.Z.; validation, D.Z. and L.W.; writing—original draft preparation, D.Z. and W.L.; writing—review and editing, D.Z. and Y.W.; project administration, Y.L.; funding acquisition, Y.L. All authors have read and agreed to the published version of the manuscript.

**Funding:** This work was supported by the National Key Research and Development Program of China (No. 2019YFB1803103).

**Data Availability Statement:** The data that support the findings of this study can be downloaded at http://moon.bao.ac.cn (accessed on 3 August 2021).

**Acknowledgments:** We are grateful to the Tianwen-1 payload team for their mission operations and to the China National Space Administration for providing the Tianwen-1 data that enabled this study. The data used in this work were processed and produced by the Ground Research and Application System (GRAS) of China's Lunar and Planetary Exploration Program, which is provided by the China National Space Administration.

**Conflicts of Interest:** The authors declare no conflict of interest.

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
