# Peer review of "Rover Attitude and Camera Parameter: Rock Measurements on Mars Surface Based on Rover Attitude and Camera Parameter for Tianwen-1 Mission"

_remotesensing, doi:10.3390/rs15184388_

Round 1
Reviewer 1 Report
Overall, this is a well-written and interesting paper that does an important job of documenting the state-of-the-art in Chinese rover data analysis. Minor suggestions and questions are included in the attached marked-up .pdf file.
I feel one point is worth repeating and emphasizing. The paper would be much stronger if some quantitative estimates of uncertainties were included. How accurately can the values in the R and t matricies be estimated? How accurate are the distances and sizes after refinement through stereogrammetry?
I also encourage the authors to look at the NAIF toolkit as it is a widely used method for doing the matrix rotations, translations, and other operations described in this paper.
Thank you

The English is of generally high quality with only a few places where some phrases should be revised.
Author Response
Dear Reviewers,
We are truly grateful for your critical comments and thoughtful suggestions. We have made careful and thorough modifications to the manuscript based on these comments and suggestions. We are genuinely thankful for your dedication in ensuring the rigor and clarity of our paper.
We have revised the manuscript (Title: RACP: Rock Measurements on Mars Surface based on Rover Attitude and Camera Parameter for Tianwen-1 Mission) and sent the revised manuscript to you. Besides, we have sent you a response file and a marked manuscript.
Your constructive critiques and recommendations have played a pivotal role in refining our research. Thank you for your invaluable contribution.
Sincerely yours,
Yu Liu on behalf of the authors.
Corresponding author: Yu Liu
E-mail: liuy@bupt.edu.cn

Reviewer 2 Report
This manuscript provides a valuable approach for practical applications on the rock measurements on Mars surface with the Navigation and Terrain Camera (NaTeCam) on the Zhurong rover. But there are still some issues that need to be considered:
(1) This paper describes some contents unrelated to the research.
(2) This paper used the data of NaTeCam on the Zhurong rover, the method is universal and can be extended to other similar tasks.
(3) This paper reviews some matching algorithms, but which method is used in your study? What is the matching process?
(4) In section 3.3, the description in the paper is inconsistent with the framework (Fig.5).
(5) It is recommended to add a data description section to describe the experimental data used in this paper, and then all the test results are unified on these image data. In addition, Figure 7,10 and 11-14 label the Sol number or image date.
(6) In Figure 14, Since the panoramic map has been stitched together and is also transferred to the Mars coordinate system according to the conversion model, it can be used as a topographic map, but the reference lines at different distances are not relatively parallel straight lines or polygonal line.
(7) Some figures in this manuscript are small and not clear enough. In addition, a scale bar and coordinate system should be added into the figures.
In addition, I am attaching the pdf with more comments that I hope will help you improving it.

Author Response

(The authors gave the same response as above.)

Round 2
Reviewer 2 Report
Thank you to the author for making revisions to the issues I am concerned about, and the manuscript has greatly improved. But there are still some issues that need to be considered:
(1) The abbreviation of words should be noted, e.g. three-dimensional. When it first appears, it should be spelled completely, and abbreviations should be used uniformly after it.
(2) In the Introduction, although the Perseverance and Curiosity rovers are similar to the Zhurong rover, it cannot be concluded that the method proposed in this paper can be demonstrated on the Perseverance and Curiosity rovers.
(3) In the Discussion, the authors discuss two errors, but the matching error of the images is also an important term, especially the texture features on Mars images are relatively monotonous.
In addition, I am attaching the pdf with more comments that I hope will help you improving it.

a little correction of language should be
Author Response
Dear Reviewers,
We sincerely appreciate once again your valuable feedback and insightful suggestions. We have revised our manuscript based on your comments. Your dedication of time and expertise to our work has undoubtedly improved its quality. Your constructive criticisms and suggestions have played a crucial role in refining our research. Thank you for your invaluable contributions.
We would like to express our heartfelt gratitude to the reviewer for their feedback, which has greatly assisted us in revising our manuscript. Below is our point-by-point response to your concerns and comments.
Sincerely yours,
Yu Liu on behalf of the authors.
Corresponding author: Yu Liu
E-mail: liuy@bupt.edu.cn
